# Epigenome-Wide Association Analysis of Differentially Methylated Signals in Blood Samples of Patients with Non-Small-Cell Lung Cancer

**DOI:** 10.3390/jcm8091307

**Published:** 2019-08-25

**Authors:** Yoonki Hong, Hye-Mi Choi, Hyun Sub Cheong, Hyoung Doo Shin, Chang Min Choi, Woo Jin Kim

**Affiliations:** 1Department of Internal Medicine, School of Medicine, Kangwon National University, Chuncheon 200-701, Korea; 2Division of Biomedical Convergence, College of Biomedical Science, and Institute of Bioscience & Biotechnology, Kangwon National University, Chuncheon 200-701, Korea; 3Department of Genetic Epidemiology, SNP Genetics, Inc., Sogang University, Seoul, 04107, Korea; 4Department of Life Science, Sogang University, Seoul 121-742, Korea; 5Department of Pulmonary and Critical Care Medicine, Asan Medical Center, University of Ulsan College of Medicine, Seoul 05505, Korea

**Keywords:** non-small-cell lung cancer, DNA methylation, biomarker

## Abstract

Lung cancer is a common form of cancer and the leading cause of cancer-related deaths worldwide. Early diagnosis using noninvasive biomarkers may play an important role in increasing the survival rate of patients with lung cancer. Biomarkers of DNA methylation in blood samples may improve the early diagnosis of lung cancer. Here, we used peripheral blood samples obtained from 150 patients diagnosed with non-small-cell lung cancer (NSCLC) and 150 healthy controls. The latter were selected by frequency matching with the 150 patients with NSCLC, based on age, sex, and smoking status. Genome-wide methylation profiles were obtained using a MethylationEPIC BeadChip Kit, which covers the 850k bp cytosine–phosphate–guanine site. This analysis showed two significant differentially methylated changes (cg12169243 [*DPH6*] and cg25429010 [*IMP3*]) associated with NSCLC in current smokers, six changes (cg09245319, cg17183999 [*USP7*], cg06366994 [*CPE*], cg24992236 [*MEG9*], cg22144719, and cg22448179 [epidermal growth factor receptor]) associated with epidermal growth factor receptor mutation in patients with adenocarcinoma, and four changes (cg25021476 [*RSL24D1*], cg04989085 [*FAM113B*], cg20905681 [*CKAP4*], and cg26379694) associated with advanced-stage NSCLC compared with stage I NSCLC. The validation of these DNA methylation changes and further research on the related genes may help develop easily accessible biomarkers for the early diagnosis or prognosis of NSCLC.

## 1. Introduction

With 2.1 million new cases and 1.8 million deaths reported in 2018, lung cancer is a common form of cancer and the leading cause of cancer-related deaths worldwide [1]. The mortality rate remains high, partly owing to the diagnosis of cancer at an advanced stage [2]. The screening of patients with early-stage disease detected through low-dose computed tomography (CT) showed a high survival rate of approximately 90% [3]. Therefore, early diagnosis using noninvasive biomarkers may play an important role in increasing the survival rate of patients with lung cancer.

Therefore, effective noninvasive screening methods are urgently required in clinical practice. Numerous recent studies have reported the importance of molecular biomarkers in this setting [4]. Thus far, p53 and the Kirsten rat sarcoma viral oncogene homolog have been proposed as potential prognostic biomarkers. Moreover, epidermal growth factor receptor (*EGFR*), human *EGFR-2*, excision repair cross-complementation group 1, ribonucleotide reductase M1, and breast cancer gene (BRCA have been proposed as prognostic and predictive biomarkers in early-stage non-small-cell lung cancer (NSCLC) [4]. However, very few of those biomarkers have been validated.

Meanwhile, several studies have described epigenetic signatures based on DNA methylation, which are associated with the outcome in early-stage NSCLC [5]. For example, p16 and cadherin 13 have been associated with a shorter time to recurrence [6], and a five-gene methylation signature (*HIST1H4F, PCDHGB6, NPBWR1, ALX1,* and *HOXA9*) was identified in patients at a high risk of recurrence [7]. However, it is unlikely that biomarkers for the early detection of lung cancer will be clinically applied in the near future [8]. In the majority of studies, samples obtained from lung tissue were investigated. Hence, for useful easily available biomarkers in clinical practice, it may be necessary to be easily accessible and detectable (e.g., in blood samples).

Recently, studies have provided evidence that methylation changes, also associated with smoking, in peripheral blood may predict lung cancer-related mortality and improve the prediction of the risk of lung cancer [9,10]. These findings have important implications in suggesting biomarkers that can be used in clinical practice. However, there are currently numerous challenges to be overcome for the clinical application of DNA methylation markers, such as the mutual mechanism of DNA methylation changes in lung tumors and blood and the effects of DNA methylation changes in nonsmokers with lung cancer.

In this article, we present the results of an epigenome-wide association methylation study based on the detection of methylation using DNA samples extracted from the blood of 150 Korean patients with lung cancer and 150 healthy controls.

## 2. Methods

### 2.1. Study Subjects and Preparation of Tissue Samples

In this study, we used blood samples obtained from the BioResource Center of Asan Medical Center (Seoul, South Korea). The samples were donated by 150 patients diagnosed with NSCLC and 150 healthy controls who had undergone health screening between 2012 and 2014. The patients included were 40–70 years of age who were diagnosed with adenocarcinoma or squamous cell carcinoma for the first time and did not have combined cancer. Patients with small-cell or mixed lung cancer, no clinical information, no stored blood, or those who disagreed with the information were excluded. For the included patients, a similar distribution of smokers and non-smokers or women and men was performed. The 150 healthy controls were selected by frequency matching with the 150 NSCLC patients based on age, sex, and smoking status. Smoking status (current, past, and nonsmokers) was self-reported in the questionnaire. Past smokers were defined as those who stopped smoking for over one year, whereas nonsmokers were defined as those with a smoking history of less than one pack throughout their life. Appropriate informed consent was obtained from the participants, and the Institutional Review Board of the Asan Medical Center (Seoul, South Korea) approved the study (IRB no. AMC IRB 2011-0883).

### 2.2. Preparation of Genomic DNA and DNA Methylation Profiling

Bisulfite conversion was performed using the EZ DNA methylation kit (Zymo Research, Irvine, CA, USA) according to the protocol provided by the manufacturer. Genome-wide methylation profiles were obtained using a MethylationEPIC BeadChip Kit, which covers the 850k bp cytosine–phosphate–guanine (CpG) site. Methylation preprocessing steps were performed using minfi [11], Enmix [12], and ComBat [13] packages of the Bioconductor project operable in R software (version 3.3.0). Preprocessing of the methylation data was performed by background correction, adjustment of probe type differences, removal of batch effects, and probe exclusion. Briefly, IDAT files were imported using minfi. Then, data preprocessing and normalization was performed using Enmix. Probes with a detection *p*-value > 0.05 and those with multimodal distribution were filtered out. Background correction was performed with the function preprocess ENmix using unused color channels as a background parameter estimate. Probe intensities were normalized using a quantile normalization method, whereas probe type bias was adjusted using the Beta-Mixture Quantile (BMIQ) method [14]. Batch effect was removed from the data using slide and subsequently subjects as covariates as they showed the strongest influence on probe methylation variability. Batch effect was removed using ComBat. Principal component analysis (PCA) was used to identify outlier samples in the methylation data.

The 850k array provides single-nucleotide resolutions of the methylation status using 866,277 probes across the chromosomes. Of these, as a quality control procedure, we excluded probes that were of low quality (i.e., detection of p < 0.01 in any sample or a bead count <3 in ≥1% of the samples), the X or Y chromosome, or potentially influenced by nearby genetic variations (e.g., single-nucleotide polymorphisms) [15]. Additionally, we removed 40,832 probes that aligned to multiple locations and probes with genetic variants overlapping the body [16,17]. The remaining 695,162 CpGs were included in the association analyses. The probe-filtering steps are summarized in Appendix A.

### 2.3. Epigenome-Wide Association Study

In order to identify differentially methylated probes (DMPs) associated with NSCLC, a logistic regression model was used with the response variable of NSCLC status and the predictor variable of methylation values. The covariates in the statistical models were age, gender, PCA, smoking (pack-years), and estimated blood cell-type proportions.

We performed additional analyses of DMPs associated with NSCLC to independently determine the effect of smoking. DMPs associated with NSCLC in each group of current smokers and nonsmokers were analyzed. In addition, DMPs associated with smoking in each group of patients with NSCLC and control subjects were also examined.

Furthermore, we used a robust linear regression model with each subtype as the response variable and methylation values as the predictor variable to evaluate the association between the methylation values and pathological/molecular subtypes (i.e., adenocarcinoma or squamous cell carcinoma, *EGFR* mutant or *EGFR* wild type).

### 2.4. Statistical Analysis

Methylation value (β) (i.e., the proportion of methylation at a given CpG site) was used for statistical analyses, which ranged from 0 (unmethylated) to 1 (methylated). We implemented a minfi-based statistical procedure of the Houseman algorithm [18] to adjust the global DNA methylation data for white-blood-cell-type heterogeneity. This approach uses raw intensity files to calculate the relative proportion of CD4+ and CD8+ T-cells, monocytes, granulocytes, B-cells, and natural killer cells. For statistical significance, we set a threshold of p < 7.2E−8 after Bonferroni’s correction (0.05/695,162 = 7.2E−8) for significant DMPs and p < 1.0E−5, an arbitrary threshold, for suggestive DMPs.

Additionally, a survival analysis was performed for significant DMPs identified in the aforementioned analyses. Methylation values were divided into low and high groups including the lower 50% and upper 50%, respectively. Kaplan–Meier curves were produced for the low-methylation group versus the high-methylation group.

## 3. Results

The baseline characteristics of patients with NSCLC and controls are summarized in Table 1. The average age was 56 years. A total of 72 never, 30 former, and 48 current smokers were included in both groups. Among patients with NSCLC, 108 and 42 patients were diagnosed with adenocarcinoma and squamous cell carcinoma, respectively. Among the 108 patients with adenocarcinoma, 70 had mutant *EGFR*.

Based on the association analysis for NSCLC, we found 11 DMPs suggestive of an association with NSCLC, after adjusting for age, gender, PCA, smoking, and estimated cell-type proportions (Appendix A). However, we could not identify DMPs significantly associated with NSCLC after Bonferroni’s correction. The epigenome-wide association plot associated with NSCLC is presented in Appendix A.

From the DMP analyses according to the smoking status, we found 58 suggestive DMPs (p < 1.0E–05) and two significant DMPs (cg12169243 [*DPH6*], p = 4.3E–08; cg25429010 [*IMP3*], p = 5.2E–08) associated with NSCLC in the group of current smokers (Table 2, Appendix A). The epigenome-wide association plot associated with NSCLC in the group of current smokers is presented in Figure 1. On the other hand, we found 19 DMPs suggestive of an association with NSCLC in the group of nonsmokers (Appendix A). The epigenome-wide association plot associated with NSCLC in the group of nonsmokers is presented in Appendix A.

In addition, we evaluated significant CpGs associated with current smoking in each group of patients with NSCLC and controls (*n* = 48 versus 48, respectively) to identify the DNA methylation changes associated with smoking in our study. The top 30 CpG sites are presented in Appendix A. Cg03636183 (*F2RL3*) and cg05934812 (*AHRR*), two established gene sites for smoking and NSCLC, were found to be significantly associated with smoking in the NSCLC patient group. The epigenome-wide association plots associated with smoking in each group of patients with NSCLC or control subjects are presented in Appendix A.

Additionally, we evaluated DMPs significantly associated with disease stage and mutant *EGFR* status. We identified 15 DMPs suggestive of an association with stage I NSCLC compared with healthy controls (Appendix A). According to the association analysis for *EGFR* mutation in patients with adenocarcinoma (*n* = 108, *EGFR* mutant = 64), we found 159 suggestive DMPs and six significant DMPs (cg09245319, cg17183999 [*USP7*], cg06366994 [*CPE*], cg24992236 [*MEG9*], cg22144719, and cg22448179 [*EGFR*]) (Table 3). From the association analysis of patients with advanced-stage NSCLC (stages IIIb and IV, *n* = 67) versus those with stage I disease (*n* = 46), we identified 58 suggestive DMPs and four significant DMPs (cg25021476 [*RSL24D1*], cg04989085 [*FAM113B*], cg20905681 [*CKAP4*], and cg26379694) (Table 4).

From the survival analysis, cg25429010 tended to be associated with longer progression-free survival in the low-methylation group versus the high-methylation group in patients with stage I and II NSCLC (*n* = 60) (hazard ratio: 1.836; p = 0.163). cg04989085 (*FAM113B*) was significantly associated with longer progression-free survival in the high-methylation group versus the low-methylation group in patients with stage IIIb and IV NSCLC (*n* = 67; hazard ratio: 0.559; p = 0.024) (Figure 2).

## 4. Discussion

In this study, we identified novel differentially methylated signals associated with NSCLC in blood DNA. In addition, DMPs significantly associated with NSCLC in current smokers, the *EGFR* mutation status in patients with adenocarcinoma, and the survival rate in patients with advanced-stage NSCLC were identified. This is the first study to use the Illumina 860k EPIC array in blood DNA obtained from patients with NSCLC. The results of the present study may help to easily determine accessible biomarkers for the early diagnosis or prognosis of NSCLC.

With the introduction of new therapies, such as molecular targeted agents or immunotherapy, the survival rate of patients with NSCLC has increased somewhat; however, the mortality rate among patients with lung cancer remains high (approximately 20%) [2,19]. Recent attempts to reduce the lung cancer-related mortality rate by low-dose CT screening continued to report major concerns regarding false-positive results, additional evaluations of indeterminate nodules, and the potential harms linked to CT radiation [20,21]. Therefore, the development of effective noninvasive screening for the early diagnosis or prediction of the prognosis of NSCLC remains an important task. Epigenetic biomarkers such as DNA methylation changes in peripheral blood may become promising tools for the detection of NSCLC in its earlier stages. Aberrant methylation changes in NSCLC have been reported in several genomic regions from various samples, such as the sputum, bronchial aspirates, and tumor tissues [6,22,23]. However, to date, there are no clinically useful DNA methylation markers. In a recent large study, it was reported that lung cancer was detectable across stages (even in stage I) using plasma cell-free DNA [24]. The present study provides a method for the diagnosis of early-stage lung cancer in easily obtainable peripheral blood samples. However, the clinical use of whole-genome bisulfite sequencing is currently limited owing to its high cost and the need for technical expertise. Therefore, an increased understanding of the DNA methylation changes in peripheral blood remains important for the development of a cost-effective and easy method for the detection of early-stage lung cancer.

Recently, in several case-control studies nested within prospective cohorts (NOWAC, MCCS, NSHDS, EPIC-Heidelberg, and EPIC-Italy cohort) [9,10,25], it has been repeatedly reported that DNA methylation changes measured in peripheral blood samples were associated with the risk and mortality of lung cancer. These reports supported the growing evidence that DNA methylation changes measurable in peripheral blood may be useful as biomarkers for determining the risk of lung cancer. Among the discovered loci, CpGs in the *F2RL3* and *AHRR* were repeatedly associated with smoking and the risk of developing lung cancer [9]. It could be hypothesized that the hypomethylation of these CpG sites may mediate the effect of tobacco on the risk of lung cancer [25]. Our study also showed that cg03636183 (*F2RL3*) and cg05934812 (*AHRR*) were significantly associated with smoking in the NSCLC patient group, but not in the control group. This finding supports the mediation effect of tobacco on the risk of lung cancer. However, there are also studies reporting contradictory results. In a multicenter study performed in Eastern and Central Europe, no association between global DNA methylation in peripheral blood and the risk of lung cancer in nonsmoking women was reported [26]. It has been reported in another study that the association between the frequency of smoking and increased DNA methylation in circulating leukocytes exhibited racial differences [27]. Further characterization of the mechanism through which methylation is linked to the risk of lung cancer in prospective studies including larger cohorts is warranted for the clinical application of blood DNA methylation markers. Moreover, it is important to identify the functional relevance of the discovered gene regions.

In our study, we identified two significant DMPs, namely, cg12169243 (*DPH6*) and cg25429010 (*IMP3*), associated with NSCLC in current smokers. *DPH6* has not been previously associated with lung cancer, and further studies are warranted to confirm the present findings. *IMP3* has been associated with advanced stage in patients with adenocarcinoma, playing an important role in tumor invasion [28]. Further research on this gene is required to develop biomarkers for the risk and prognosis of lung cancer. We also identified six DMPs significantly associated with *EGFR* mutation in patients with adenocarcinoma. Although the five associated loci have not been associated with *EGFR* mutation, cg22448179 was identified on the *EGFR* gene. This result suggests that there is a change in the *EGFR* of white blood cells, following changes in the *EGFR* mutation in NSCLC. Further research is warranted to elucidate the specific mechanism. We also observed four DMPs significantly associated with the survival rate in patients with advanced-stage NSCLC. Among them, *CKAP4* has been suggested as a diagnostic marker for NSCLC [29]. In the study, the levels of *CKAP4* in the serum were higher in patients with stage I NSCLC compared to those observed in controls. Considering the results of our study, *CKAP4* is expected to be an important target in the development of biomarkers for the early diagnosis and prediction of prognosis of NSCLC. Despite being not a well-studied gene region, cg04989085 (*FAM113B*) was associated with advanced-stage disease compared with stage I NSCLC, as well as longer survival in patients with advanced-stage NSCLC. This region is expected to be an important target in the discovery of biomarkers for the prediction of poor prognosis in NSCLC.

We could not directly investigate the functional relevance of the genes identified in this study. To indirectly determine whether observed methylation changes affect the expression of proximal genes, for the five probes (cg12169243 in *DPH6*; cg25429010 in *IMP3*; cg25021476 in *RSL24D1*; cg04989085 in *FAM113B*; and cg20905681 in *CKAP4*), we analyzed the correlation between methylation and expression (RNA sequencing) using The Cancer Genome Atlas (TCGA, http://cancergenome.nih.gov/) data. Excluding two with no TCGA data and one with no available data, only data for *IMP3*-probe and *FAM113B* were available. *IMP3*-probe methylation suggested a weak correlation with *IMP3* expression from TCGA (Pearson’s correlation coefficient = 0.167, *p* value = 0.025 in 5’UTR and Pearson’s correlation coefficient = 0.172, *p* value = 50.15 in the first exon, respectively) (Appendix A). *FAM113B*-probe methylation may be inversely correlated with *FAM113B* expression (Pearson’s correlation coefficient = −0.581 in gene body, *p*-value < 0.01) (Appendix A).

Our study was characterized by limitations. Firstly, this was a cross-sectional study that could not guarantee the causal relationship between methylation changes and NSCLC or related outcomes. It is not possible to ascertain whether the identified DMPs have predated the development of NSCLC. Secondly, we could not replicate or validate the present findings in different populations. However, the results obtained using the 850k BeadChip array may be more meaningful for future validation studies. Finally, our results are not supported by a functional study. The gene expression profiles in lung tissues strengthen the biologic evidence that the DNA methylation alterations observed in blood are related to NSCLC, thus providing a kind of functional replication.

## 5. Conclusions

This genome-wide DNA methylation analysis identified two significant differentially methylated changes associated with NSCLC, six changes associated with EGFR mutation, and four changes associated with advanced stage in peripheral blood samples obtained from patients with NSCLC. The validation of these DNA methylation changes and further research on the related genes may help to develop easily accessible biomarkers for the early diagnosis or prognosis of NSCLC.

## Figures and Tables

**Figure 1 jcm-08-01307-f001:**
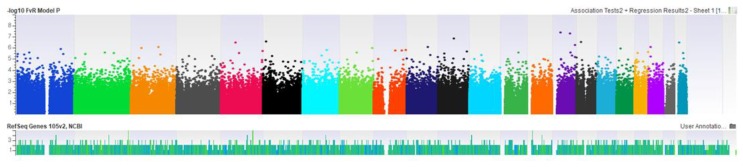
Epigenome-wide association plot associated with NSCLC in the group of current smokers. Adjusted for age, sex, smoking (pack-years), blood cell type, and PCA.

**Figure 2 jcm-08-01307-f002:**
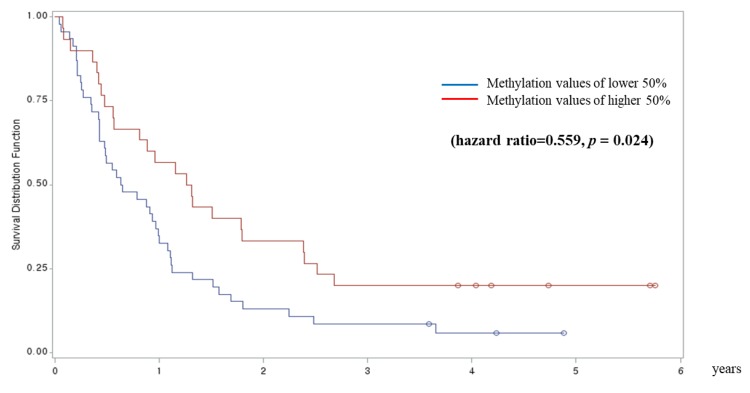
cg04989085 (*FAM113B*) is associated with longer progression-free survival in the high-methylation group versus the low-methylation group in patients with stage IIIb and IV non-small cell lung cancer.

**Table 1 jcm-08-01307-t001:** Baseline characteristics of patients with non-small-cell lung cancer (NSCLC) and matched control subjects.

	Patients	Controls
Total number	150	150
Male (%)	86 (57.3)	86 (57.3)
Age, years (mean ± standard error)	56.1 ± 7.95	55.6 ± 7.94
Smoking state (current/past/nonsmokers)	48/30/72	48/30/72
Histological type, N (%)		
Adenocarcinoma (% of EGFR mutant)	108 (59.3)	
Squamous cell carcinoma	42	
Stage (I/II/III/IV)	46/14/31/59	—

Values are presented as a number (range or %) or median with range. EGFR: epidermal growth factor receptor.

**Table 2 jcm-08-01307-t002:** Differentially methylated probes identified as significantly associated with non-small-cell lung cancer in current smokers.

Target ID	Chr ^a^	Position	Gene ID	Location	Beta ^b^	Beta (SE)	p-Value ^c^
cg12169243	15	35825579	*DPH6*	Body	127.250	30.577	4.3E–08
cg25429010	15	75932582	*IMP3*	5′UTR	−529.523	128.213	5.2E–08

^a^ Chromosome; ^b^ Regression coefficient from the statistical models. The covariates age, sex, principal component analysis (PCA), smoking (pack-years), and estimated cell-type proportions were included in the models. Methylation (beta) ranged between 0 and 1 in the analyses; ^c^ Statistical significance from the statistical model.

**Table 3 jcm-08-01307-t003:** Differentially methylated probes significantly associated with the *EGFR* status in patients with adenocarcinoma of NSCLC.

Target ID	Chr ^a^	Position	Gene	Location	Beta ^b^	Beta (SE)	p-Value ^c^
cg09245319	2	227049703	—	—	−227.7	66.0	3.5E–09
cg17183999	16	8998406	*USP7*	Body	−433.8	124.7	1.3E–08
cg06366994	4	166298963	*CPE*	TSS1500	−51.9	13.3	2.0E–08
cg24992236	14	101536808	*MEG9*	Body	133.9	34.9	2.2E–08
cg22144719	9	138106031	*—*	—	−55.5	14.0	2.2E–08
cg22448179	7	55166882	*EGFR*	Body	−93.7	26.3	7.2E–08

^a^ Chromosome; ^b^ Regression coefficient from the statistical models. The covariates age, sex, PCA, smoking (pack-years), and estimated cell-type proportions were included in the models. Methylation (beta) ranged between 0 and 1 in the analyses; ^c^ Statistical significance from the statistical model.

**Table 4 jcm-08-01307-t004:** Differentially methylated probes significantly associated with advanced stage in patients with NSCLC.

Target ID	Chr ^a^	Position	Gene	Location	Beta ^b^	Beta (SE)	p-Value ^c^
cg25021476	15	55488413	*RSL24D1*	Body	−771.0	182.1	3.9E–08
cg04989085	12	47629182	*FAM113B*	Body	190.3	47.0	8.7E–08
cg20905681	12	106641994	*CKAP4*	TSS1500	905.8	228.2	1.3E–07
cg26379694	5	152732350	—	—	35.4	9.0	1.3E–07

^a^ Chromosome; ^b^ Regression coefficient from the statistical models. The covariates age, sex, PCA, smoking (pack-years), and estimated cell-type proportions were included in the models. Methylation (beta) ranged between 0 and 1 in the analyses; ^c^ Statistical significance from the statistical model.

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
