# Peer review of "Epigenome-Wide Association Analysis of Differentially Methylated Signals in Blood Samples of Patients with Non-Small-Cell Lung Cancer"

_jcm, 2019, doi:10.3390/jcm8091307_

Round 1
Reviewer 1 Report
This is an epigenome-wide association methylation study on Korean population based on the detection of methylation using DNA samples extracted from the blood of 150 patients with non-small cell lung cancer (NSCLC) and 150 healthy controls. This study used the Illumina 860k EPIC array in blood DNA obtained from patients with lung cancer. Authors have investigated differentially methylated signals associated with NSCLC, differentially methylated probes (DMPs) significantly associated with NSCLC in current smokers, the EGFR mutation status in patients with adenocarcinoma, and the survival rate in patients with advanced stage NSCLC.
The results of this genome-wide DNA methylation analysis showed two significant differentially methylated changes associated with NSCLC, six changes associated with EGFR mutation, and four changes associated with advanced stage in peripheral blood samples obtained from patients with NSCLC. This study is significantly interesting as identification of epigenetic biomarkers such as DNA methylation changes in blood may become promising tools for early detection of NSCLC.
Overall, the manuscript is written well, methodology used in the study seems appropriate, the results are explained and discussed well. The main figures are supported with various supplementary tables and figures. However, some minor corrections and suggestions in the manuscript are provided below.
Authors may provide inclusion and exclusion criteria for NSCLC patients utilized in this study in the methodology section. Statistical analysis section needs to be elaborated with more information such as statistical program/ tools/ software and parameters used in the study. A separate heading for statistical analyses section would be better for clarity.Author Response
We greatly appreciate your valuable comments. We have carefully revised and rechecked the manuscript.
We have added the inclusion and exclusion criteria for NSCLC patients in the Methods section (section 2.1)
We have revised and added the methods (section 2.2) and have prepared a separate heading for statistical analyses (section 2.4).
Reviewer 2 Report
In their study, the authors looked for biomarkers of DNA methylation in peripheral blood samples from 150 patients with NSCLC and 150 healthy controls. The authors identified two markers associated with NSCLC in current smokers, 6 associated with EGFR mutation in patients with adenocarcinoma, and 4 associated with high vs low stage in NSCLC. This study is the first to use the Illumina 860k EPIC array in blood DNA obtained from patients with NSCLC. In addition, blood screen, as mentioned by the authors, presents the advantage to be non-invasive, compared to current screening methods for early cancer detection.
While the article is very well written and highlights the importance of the research, authors should address the following comment:
The complexity of this type of studies that doesn’t validate the function of genes identified in cancer, but still remain a crucial starting point, is to reach a large audience that is not necessarily a specialist in methylation analysis or bioinformatics. I would personally encourage the authors to develop a bit more the discussion and maybe some parts of the result section (when possible) and describe more what is known about the function of the genes identified 1-in lung cancer, but also 2- in other cancers to highlight their potential relevance as biomarkers.
Author Response
We deeply appreciate your valuable comments.
To describe more about what is known about the function of the genes identified, we have added information regarding the correlation of methylation and expression using available public data.